METHODS

# NeuroMotion: Open-source platform with neuromechanical and deep network modules to generate surface EMG signals during voluntary movement

Shihan Ma[1,2], Irene Mendez Guerra[1], Arnault Hubert Caillet[1], Jiamin Zhao[2], Alexander Kenneth Clarke[1], Kostiantyn Maksymenko[3], Samuel Deslauriers-Gauthier[3,4], Xinjun Sheng[2,5], Xiangyang Zhu[2,5], Dario Farina[1] *

1 Department of Bioengineering, Imperial College London, London, United Kingdom, 2 State Key Laboratory of Mechanical System and Vibration, Shanghai Jiao Tong University, Shanghai, China, 3 Neurodec, Sophia Antipolis, France, 4 Inria Centre at Université Côte d'Azur, Nice, France, 5 Meta Robotics Institute, Shanghai Jiao Tong University, Shanghai, China

* d.farina@imperial.ac.uk

**Data Availability Statement:** All programming code are publicly available on GitHub, at https://github.com/shihan-ma/NeuroMotion under the

## Abstract

Neuromechanical studies investigate how the nervous system interacts with the musculo-skeletal (MSK) system to generate volitional movements. Such studies have been supported by simulation models that provide insights into variables that cannot be measured experimentally and allow a large number of conditions to be tested before the experimental analysis. However, current simulation models of electromyography (EMG), a core physiological signal in neuromechanical analyses, remain either limited in accuracy and conditions or are computationally heavy to apply. Here, we provide a computational platform to enable future work to overcome these limitations by presenting NeuroMotion, an open-source simulator that can modularly test a variety of approaches to the full-spectrum synthesis of EMG signals during voluntary movements. We demonstrate NeuroMotion using three sample modules. The first module is an upper-limb MSK model with OpenSim API to estimate the muscle fibre lengths and muscle activations during movements. The second module is Bio-Mime, a deep neural network-based EMG generator that receives nonstationary physiological parameter inputs, like the afore-estimated muscle fibre lengths, and efficiently outputs motor unit action potentials (MUAPs). The third module is a motor unit pool model that transforms the muscle activations into discharge timings of motor units. The discharge timings are convolved with the output of BioMime to simulate EMG signals during the movement. We first show how MUAP waveforms change during different levels of physiological parameter variations and different movements. We then show that the synthetic EMG signals during two-degree-of-freedom hand and wrist movements can be used to augment experimental data for regressing joint angles. Ridge regressors trained on the synthetic dataset were directly used to predict joint angles from experimental data. In this way, Neuro-Motion was able to generate full-spectrum EMG for the first use-case of human forearm electrophysiology during voluntary hand, wrist, and forearm movements. All intermediate variables are available, which allows the user to study cause-effect relationships in the

**Funding:** This work was supported by the European Research Council (ERC) under the Synergy Grant Natural BionicS (810346 to DF), the Engineering & Physical Sciences Research Council (EPSRC) Transformative Healthcare for 2050 project Non-Invasive Single Neuron Electrical Monitoring (NISNEM Technology, grant EP/T020970/1 to DF), and by the National Natural Science Foundation of China (52227808 to XZ). The funders had no role in study design, data collection and analysis, decision to publish, or preparation of the manuscript.

**Competing interests:** • I have read the journal's policy and the authors of this manuscript have the following competing interests: Kostiantyn Maksymenko and Samuel Deslauriers-Gauthier are the co-founders of the company Neurodec; all other authors have no competing interests.

complex neuromechanical system, fast iterate algorithms before collecting experimental data, and validate algorithms that estimate non-measurable parameters in experiments. We expect this modular platform will enable validation of generative EMG models, complement experimental approaches and empower neuromechanical research.

## Author summary

Neuromechanical studies investigate how the nervous system and musculoskeletal system interact to generate movements. Such studies heavily rely on simulation models, which provide non-measurable variables to complement the experimental analyses. However, the simulation models of surface electromyography (EMG), the core physiological signal widely used in neuromechanical analyses, are limited to static conditions. We bridged this gap by proposing NeuroMotion, the first full-spectrum EMG simulator that can be used to generate EMG signals during voluntary movements. NeuroMotion integrates a musculoskeletal model, a neural network-based EMG generator, and advanced motoneuron models. With representative applications of this simulator, we show that it can be used to investigate the variabilities of EMG signals during voluntary movement. We also demonstrate that the synthetic signals generated by NeuroMotion can be used to augment experimental data for regressing joint angles. We expect the functionality provided by NeuroMotion, which is provided open-source, will stimulate progress in neuromechanics.

## Introduction

Human neuromechanics is the discipline that combines neuroscience and biomechanics to reach a fundamental understanding of the interactions between the nervous, muscular, and skeletal systems during human movements [1]. It allows us to uncover the functions and mechanisms of the nervous system under the production of movements [2, 3] by studying the movements from the perspective of their neural control [4]. Neuromechanical investigations are also important for developing decoding methods by providing the link between neural activities and behaviours. The decoding methods can be used to identify human intent from neural signals generated during a specific behaviour (neural interfaces). Neuromechanical studies have therefore allowed us to address problems ranging from motor control [4], rehabilitation engineering [5], and human-machine interfaces [6].

Due to the complex interplay of the nervous, muscular, and skeletal systems, researchers have used multiple simulation tools to explore certain aspects of human neuromechanics. A series of models that describe the neuron structures and functions have been proposed, including the classical Hodgkin-Huxley model [7]. At a macro scale, several analytical and numerical electromyogram (EMG) models have been used to study the electrical outputs produced by skeletal muscles upon activations of populations of neurons [8–11]. In parallel, multiple simulation platforms, such as OpenSim [12, 13] and MuJoCo [14, 15], have been widely used to study human body movements during dynamic simulations in biomechanical studies. However, the electrical outputs are rarely simulated together with the biomechanical system during voluntary movements.

OpenSim allows the users to include EMG signals as an additional input for better estimating the musculo-tendon dynamics and parameters but not to forwardly generate EMG signals during a movement [12]. Fuglevand's model has been widely used to study muscle force,

motoneuron activities, and EMG signals but is limited to isometric contractions [16]. A recurrent neural network has been recently used as a black box to convert motions to the down-sampled and smoothed muscle activities, without knowing the internal parameters of the system [17]. This inhibits using the model to study cause-effect relations between system parameters or to validate algorithms that require internal information. For instance, validating adaptive decomposition algorithms requires knowing the firing status of each motoneuron. One primary reason for the lack of an integrated and precise EMG simulator feasible for voluntary movements is the absence of models that link EMG generation to movement biomechanics. Another reason is that current advanced EMG models are not efficient enough to adapt to the non-stationary physiological parameters during voluntary movements. These two challenges have been addressed by our recently released EMG model, BioMime [18], which is a conditional generative model that takes the physiological parameters as inputs and outputs the dynamic motor unit action potential (MUAP) signals efficiently.

Using BioMime as a key module, here we propose NeuroMotion, an open-source EMG generative model that replicates the full-spectrum generation of electrophysiological signals during voluntary movements. EMG signals are produced by the mixed convolutions of the MUAPs (electrical potential templates generated when one motoneuron is activated) and spike trains (continuous discharge timings of a motoneuron) of all active motor units during a movement. Correspondingly, NeuroMotion consists of three modules. The first module is an upper-limb musculoskeletal (MSK) model with OpenSim API for defining and visualising the movement and estimating the muscle fibre lengths and muscle activations during the movement. The muscle fibre lengths are then utilised by the second module, BioMime, to simulate the dynamic MUAPs during the movement. The third module is a motor unit pool model that receives the neural inputs derived from the muscle activations and outputs stimulations to each muscle in the format of spike trains. An overview of NeuroMotion is shown in Fig 1. With NeuroMotion, users can synthesise large EMG datasets under a vast repertoire of hand and wrist movements with all intermediate variables available. One potential application is to fast iterate regression and classification algorithms on a synthetic dataset before collecting experimental data (this data augmentation example will be used later in the Results). Another application is the validation of information extraction algorithms (e.g., EMG decomposition algorithms) when experimental data lacks the ground truth. Data, codes, and instructions are available at https://github.com/shihan-ma/NeuroMotion.

## Methods

### Overview

NeuroMotion is a surface EMG generative model that is designed to simulate EMG signals during voluntary human upper limb movements. NeuroMotion takes the kinematics of human hand, wrist, and forearm as inputs and outputs the synthetic surface EMG signals that explain the myoelectric activities responsible for the movement. Three modules are incorporated to complete this full-spectrum simulation, including an upper-limb MSK model with OpenSim API, BioMime, and a Motor Unit Pool model. Intermediate variables (changes in physiological parameters, MUAPs, and spike trains) are available in this hierarchical and modular simulation. Details of the three key modules are described in Section Core modules. An overview of the workflow of NeuroMotion is shown in Fig 1.

### Core modules

**MSK model with OpenSim.** OpenSim is an open-source software platform that is used for modelling and analysing neuromusculoskeletal systems [12, 13]. With this platform,

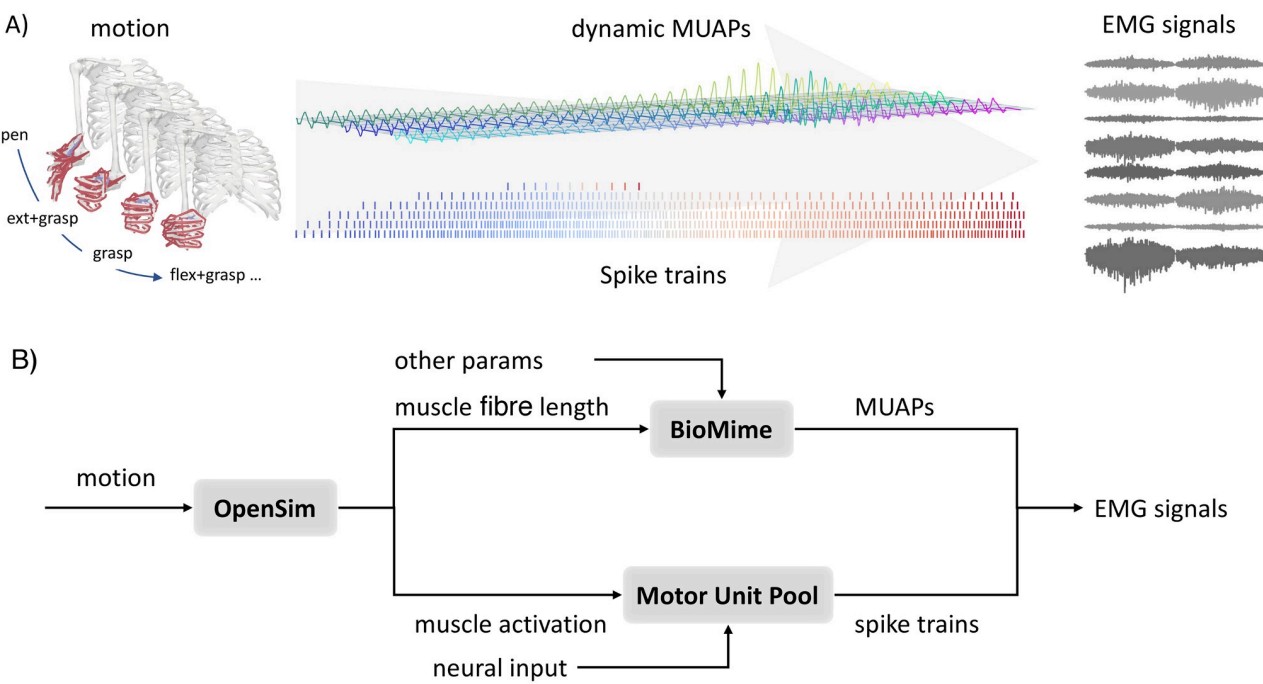

**Fig 1. Overview of NeuroMotion.** A) NeuroMotion provides an integrated generative model to simulate forearm surface EMG signals during voluntary hand and wrist movements. B) The movements are defined by the kinematics of an upper limb MSK model. Muscle fibre lengths during the movement are estimated by **OpenSim** and imported into **BioMime** to simulate dynamic MUAPs, which are the continuously changing electrical potential templates generated by the activations of the motor units. The other physiological parameters, including the current source propagation velocity and motor unit depth, can be estimated from muscle fibre length and imported to BioMime as well. Neural inputs to each muscle, which can be derived from the normalised muscle activations from OpenSim or set by the users, are further transformed into spike trains by the **Motor Unit Pool** model. Surface EMG signals are finally calculated as the summation of the convolution of the dynamic MUAPs and the spike trains among all active motor units.

researchers can build MSK models, simulate dynamic movements, and perform motion analysis for biomechanical studies that advance movement science. Here, we use OpenSim as a bridge between movements and physiological system states for three purposes: (1) to define and visualise movements of MSK models, (2) to track the changes in muscle fibre lengths during the movement, and (3) to predict the muscle activations across muscles.

Since we mainly focus on simulating EMG signals from the human forearm during hand, wrist, and forearm movements, we chose the ARMs Wrist and Hand Model [19] (ARMs in abbreviation) from the large database of opens-source OpenSim models as the MSK model used in NeuroMotion. The original ARMs model includes 23 degrees of freedom (DoFs), including the full DoFs of finger movements and the flexion/extension and radial/ulnar deviation DoFs for the wrist. We added the pronation/supination DoF to the source files of ARMs model such that a full range of hand, wrist, and forearm movements could be simulated.

The MSK model in NeuroMotion can be customised to the subject's anthropometry in three ways. First, a fully personalised MSK model can be created by acquiring medical images of the MSK system and building physics-based models. This process requires considerable skills and manual interventions, but has been facilitated by the advancement of automatic tools [20, 21] and associated machine learning methods [22]. A more practical approach, which is automated in OpenSim, is to scale each segment in the ARMs model by minimising the distances between the virtual markers placed on ARMs model and the real markers placed on the subject. The ARMs model can also be manually scaled by anthropometric measurements.

The movement defined by the user can be visualised and investigated in the OpenSim GUI. Muscle-tendon lengths during the movement are related to the joint angles and are tracked and extracted using the OpenSim built-in functions. Assuming tendons to be rigid at constant slack length, the muscle fibre lengths are deduced and imported into BioMime to generate the dynamic MUAPs. Furthermore, the individual muscle activations responsible for the movement are estimated with the built-in Static Optimization tool in OpenSim. Static Optimisation solves the inverse dynamics by minimising the cost function of muscle activations under the constraints of muscle activation-to-force conditions. The exported muscle activations are further normalised and imported into the Motor Unit Pool model to act as the neural input to the motoneuron pools, according to the linearity properties of the motoneuron pools [23]. OpenSim's built-in tools have been integrated into NeuroMotion by its Python API.

**BioMime.** After OpenSim captures the changes in a biomechanical system from the joint kinematics, BioMime is used to simulate the corresponding MUAPs during the movement. BioMime is a deep conditional generative model that simulates electrical potential fields given a set of physiological parameters [18]. A schematic of BioMime's training and inference pipeline is shown in Fig 2. Learning from the outputs of its teacher numerical model [10], BioMime captures the relations between the physiological parameters and the MUAP templates and essentially replicates the biophysical properties of the volume conductor of the forearm. When the time-varying parameters during a movement are imported into BioMime, the output will change accordingly. Therefore, BioMime can be used to generate a sequence of MUAP templates during any movement as long as the parameter changes are available and plausible. The computational cost to simulate a movement in high temporal resolution is extremely low given BioMime's ultra-fast inference speed (0.287 seconds per muscle per condition [18]).

The public BioMime model was trained on a dataset generated by a numerical model with a specific forearm anatomy. Therefore, the BioMime model can be regarded as a subject-specific surrogate model of the volume conductor with this forearm anatomy. The user can keep this well-trained model or train their individual BioMime. Detailed instructions on how to train BioMime can be found in https://github.com/shihan-ma/BioMime. The newest version of BioMime supports changes in seven physiological parameters, including the conductivity of the fat layer and the fibre number, depth and medial-lateral position, innervation zone, conduction velocity, and fibre length of each motor unit (Details in the supplementary of [18]). Some of these parameters can be measured experimentally. For example, the location and the fibre length of a motor unit could be estimated by using ultrasound [24, 25]. The location of the innervation zone and the muscle fibre conduction velocity can be estimated, for example by using high-density EMG recordings [26, 27]. Some parameters can also be obtained from the

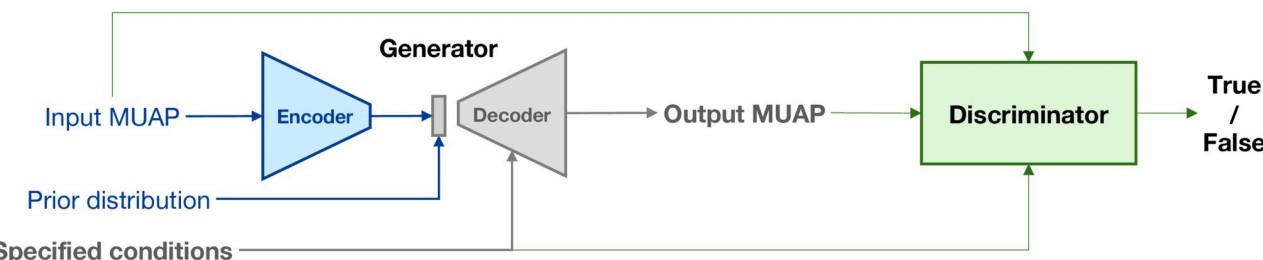

**Fig 2. Schematic of BioMime.** BioMime is a conditional generative model trained in an adversarial way. The generator takes the specified conditions and latent features of MUAPs as the inputs and outputs the simulated MUAP signals. The latent features can be encoded from existing MUAPs or sampled from a prior distribution. The discriminator distinguishes real samples from fake samples conditioned on the specified parameters. Only the generator is used in NeuroMotion.

synthetic movement of the MSK model. As described in Section MSK model with OpenSim, muscle fibre length can be estimated by OpenSim. The depth of a motor unit territory and the motor unit conduction velocity can be approximated from the muscle fibre length under some assumptions (see Section Estimate parameter changes).

As a conditional generative model trained in an adversarial manner, BioMime has an encoder-decoder structured generator and a discriminator. During the inference, only the generator is required. BioMime can be used to generate new MUAPs by morphing an existing MUAP or by sampling from a prior distribution. The difference is that by morphing an existing MUAP, the output MUAPs keep the properties of the input MUAP that are separated from the seven physiological parameters, while these properties are sampled from a prior distribution by sampling. When these latent properties are obtained, MUAPs that change under dynamic movements can be simulated by passing the latent and the changing physiological parameters into the decoder. Since the seven parameters explain the bulk of variations of the MUAP templates of the specific subject [8, 9, 28], the difference between the MUAPs generated by morphing or by sampling is small. NeuroMotion supports both of these two strategies and provides a dataset of MUAPs paired with their physiological parameters for generation under the morphing pattern.

**Motor unit pool.** The purpose of including the motor unit pool model in NeuroMotion is twofold: (1) to initialise the properties of the motor units within one muscle, and (2) to simulate the motoneuron activities and generate their spike trains. In NeuroMotion, we implemented two types of motor unit pool models, the classical Fuglevand's model proposed in [16] and a cohort of leaky fire-and-integrate (LIF) neuron models adapted from [29].

The properties of the motor units in the two models are initialised following the rules below. In both models, the motor unit size (number of muscle fibres within each unit) is proportional to the amplitude of the motor unit twitch force [16, 30], and the summation of the motor unit sizes meets the expected total number of fibres in one muscle (Eq 1). In the classical model, the peak twitch force is exponentially distributed while in the LIF-based model, the twitches are distributed following the linear-exponential function in Eq 2. Eq 2 was derived from experimental measurements in the literature on human forearm muscles [16, 31, 32] following the method described in [29]. The total number of fibres in one muscle was estimated as the ratio between muscle physiological cross-sectional area and the average fibre cross-sectional area for typical human forearm muscles [16, 33]. In the classical Fuglevand's model, the conduction velocity is normally distributed within the common range and then sorted from small to large to be positively related to the motor unit size. In the LIF-based model, we adapted the mathematical relationships between the axonal conduction velocity and the neuron surface area from [34] to estimate the fibre conduction velocity. In both models, the depth and medial-lateral positions of motor units are uniformly distributed within the muscle territories. The innervation zone and fibre length are normally distributed within the common ranges.

$$MN_{size} = \frac{\overline{f^{tw}}(j)}{\sum_{k=1}^{N} \overline{f^{tw}}(k)} \cdot N_f, j \in [1, N] \tag{1}$$

where $\overline{f^{tw}}(j)$ is the estimated twitch force of the $j$th motor unit in the population, $N$ the number of motor units, and $N_f$ the total number of fibres in one muscle.

$$\overline{f^{tw}}(j) = 0.81 \cdot (18.51 \cdot \frac{j}{N} + 104.10^{\frac{j^{4.83}}{N}}), j \in [1, N] \tag{2}$$

In both models, the motoneurons innervating a muscle are sequentially recruited in terms of the intensity of the neural input. The smallest motor unit is first recruited with the lowest impulse response amplitudes. Once recruited, the motoneuron starts to fire regularly. The firing rates of the recruited motor units increase with the input to the motoneuron pool until the peak firing rates are reached. The two models both predict the specific discharge behaviour (spike trains) of the motoneurons from the intensity of the neural input. Specifically, the firing state of a motor unit is decided by the recruitment threshold of the motor unit in the classical Fuglevand's model [16]. The recruitment thresholds are exponentially distributed such that few motor units have high thresholds and many motor units have low thresholds. In the LIF model-based approach, each LIF model captures the complex nonlinear MN dynamics by using a parallel combination of a leaky resistor and a capacitor. A spike is generated when the transmembrane voltage reaches a certain threshold, followed by a refractory during the action potential. With physiologically realistic distributions of the electrophysiological properties and a further interpolation between the motoneuron-specific electrophysiological properties supported by experimental data [29] in the LIF model, the predictions of both models are consistent with the onion skin theory [35] and Henneman's size principle of sequential motoneuron recruitment [36].

## Toolbox functions

In this section, we introduce the basic functions provided by NeuroMotion to define the movement of the ARMs model (Section Define movement), track the changes in physiological parameters (Section Estimate parameter changes), set common drives to motoneuron pools (Section Set neural inputs to motoneuron pools), and define the structure of motor unit pools (Section Configure motor unit pools). Python code examples for each function are displayed.

**Define movement.** Movements are simulated by using the ARMs model in NeuroMotion. A python class **MSKModel** was implemented to handle the related functions. NeuroMotion supports three approaches to defining the movement of the ARMs model. The most basic way is by assigning joint angles to each DoF by using the function *'load_mov'* (Fig 3A). The joint angles can be obtained from motion capture data (like in [37]) or be measured from sensors (as by data glove and angle sensors). There are 24 DoFs in the ARMs model and each can be changed separately within a predefined range. NeuroMotion automatically checks whether each input joint angle is within the correct range. It is possible that a movement defined by a set of joint angles is not feasible even though each joint angle is reasonably assigned. For example, a combination of joint angles may result in a sudden and unlikely change in muscle fibre lengths. We encourage the users to visualise the movement in OpenSim GUI and to check if there are such aberrations. The motion file required by OpenSim can be generated from joint angles by using the function *'write_mov'* in NeuroMotion (Python code example at Line 10 in Code Block 1).

The second way to define a movement is by interpolating between predefined poses. NeuroMotion provides eight default poses, including hand open/grasp, wrist flexion/extension, wrist radial/ulnar deviation, and forearm pronation/supination (Fig 3B). Users can define their customised poses by creating new poses in OpenSim GUI. A movement can be simulated by setting one pose at each stage from the predefined poses and setting the durations of each transition (Python code example at Lines 1-5 in Code Block 1). The eight default poses can be combined before interpolation by concatenating the names of the poses, e.g., 'open+flex' means hand open and wrist flexion at the same time. The joint angles are summed after the combination. Users may want to define the movement directly from the motion capture data,

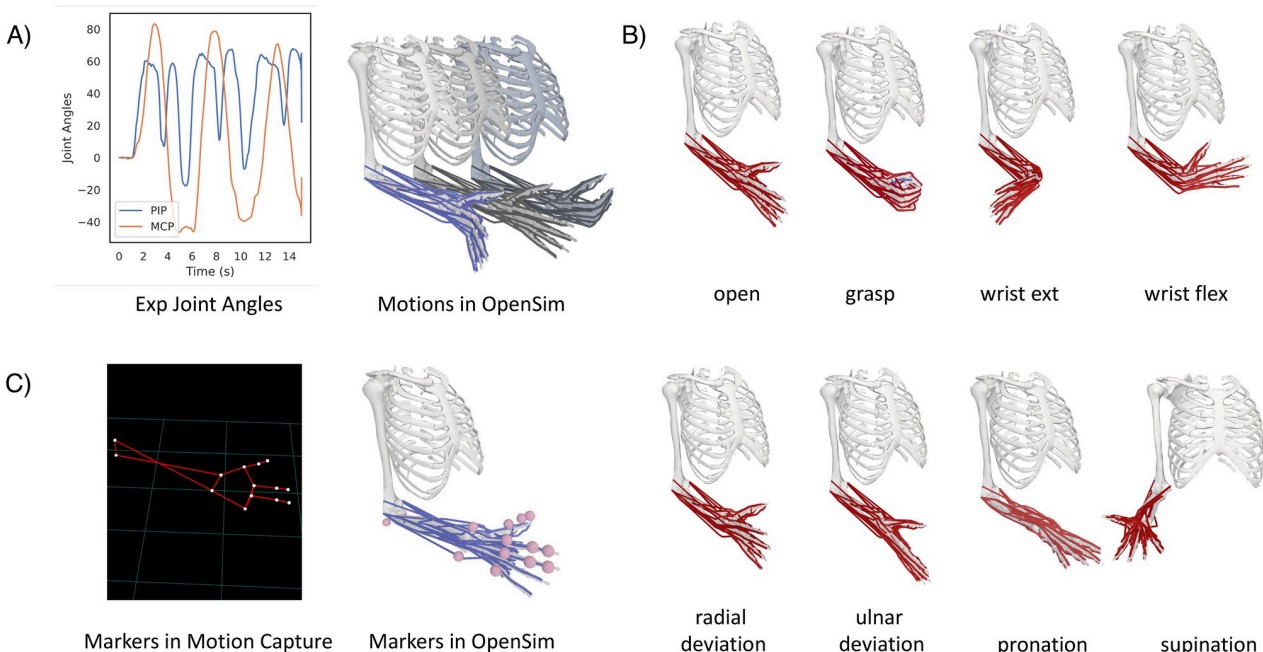

**Fig 3. Illustration of the function of movement definition in NeuroMotion.** The MSK models are driven by the joint angles in OpenSim. NeuroMotion provides three ways to define a movement. A) The most basic way to define a movement is by setting the joint angles directly. In this example, the ARMs model is driven by the metacarpophalangeal (MCP) and proximal interphalangeal (PIP) joints of the five digits. B) Eight default poses are provided. Users can combine these poses and interpolate between them to simulate a smooth movement. C) Motion capture data can be used to drive the MSK model. Markers should be aligned between experimental and simulation settings.

as the third way displayed in Fig 3C. This is possible with OpenSim GUI. The users need to place virtual markers at the same anatomical positions on the MSK model as the experimental positions. Then the MSK model can be scaled and the movement can be driven by the motion capture data.

```
1    # build msk model and simulate a movement
2    msk = MSKModel()
3    poses = ['default', 'default+flex', 'default']
4    durations = [2] * 2    # 2 seconds for each transition
5    fs = 5    # 5 Hz
6    ms_labels = ['ECRB', 'ECRL', 'PL', 'FCU', 'ECU', 'EDCI', 'FDSI']
7    msk.sim_mov(fs, poses, durations)
8    ms_lens = msk.mov2len(ms_labels=ms_labels)
9    changes = msk.len2params()
10   msk.write_mov('./res/mov.mot')
```

**Code Block 1.** Example python codes for defining a wrist flexion movement by interpolation and for acquiring changes in physiological parameters.

**Estimate parameter changes.** BioMime can simulate MUAPs during the changes in seven physiological parameters. Users can import the parameters measured during an experiment. If these physiological parameters are not available from the experiment, NeuroMotion provides an alternative method to approximate some of the parameters. Muscle fibre lengths during a movement of the MSK model can be estimated by using the Muscle Analysis Tool in

OpenSim. Two more parameters can be approximated from the muscle fibre length changes. Under the assumption of constant muscle volumes during a movement [11], the cross-sectional area of a muscle changes inversely with the muscle fibre length. Then the conduction velocity can be tracked since it is positively correlated with the cross-sectional area of muscle volumes. If the location of a motor unit consistently changes with the radius of the cross-section, the depth of the motor unit can be estimated as well. Therefore, changes in three physiological parameters can be estimated during a movement. These changes can be obtained by using the function of *'len2params'* in NeuroMotion (Python code example at Line 9 in Code Block 1).

**Set neural inputs to motoneuron pools.** Activations that trigger muscle contractions are often represented by spike trains. The spike trains are generated by motoneurons by transforming the input they receive and are convolved with the MUAP templates to generate the interference EMG signals. It still remains a challenge to decode the real neural input to motoneurons from experimental EMG data, and thus it is difficult to set these inputs to their true values during a movement.

NeuroMotion provides three ways to set the input to each motoneuron pool. First, the user can set predefined and commonly used activation profiles, which include constant, trapezoidal, triangular, and sinusoidal activations. The amplitude and duration of each type of activation can be easily changed. Second, as described in Section MSK model with OpenSim, muscle activations during a movement can be estimated by OpenSim with the Static Optimisation Tool. The muscle activations are then normalised to the activations during the maximum voluntary contractions and used as the neural inputs to the muscles. Third, NeuroMotion provides the interfaces for the user to set the muscle activations by the outputs of their algorithms, for example, using the normalised EMG signals recorded during experiments or the signals derived from muscle synergies [38].

**Configure motor unit pools.** NeuroMotion organises the motor unit pool structures by providing a python class *MotoneuronPool*. This class parameterises the electrical and mechanical properties of a motor unit pool, of which each property can be easily customised. Fundamental properties include the motor unit size, conduction velocity, motor unit positions, innervation zones, and fibre lengths. In the implementation of classical Fuglevand's model, the motor unit recruitment threshold, the minimum and maximum firing rate of each motor unit, variability of the inter-pulse intervals, and motor unit twitch force and contraction time can be manually assigned. In the LIF-based model, the motoneuron-specific parameters, such as the resistance, capacity, and time constant, are set given the mathematical descriptions in [29]. Example Python codes that define a classical motor unit pool model, define the input to motoneurons and calculate the spike trains are shown in Code Block 2.

Users may prefer using the classical Fuglevand's model as it is more conceptually simple with fewer parameters. The LIF-based model might be preferred as it provides a more physiological way to predict the discharge activities based on the physiological dynamics of motoneuron membrane charging and discharging and allows a finer control of motoneuron properties by providing more controllable parameters. Users could also add their self-developed models, for example the Izhikevich neuron model [39], into NeuroMotion.

## Results

NeuroMotion provides the input, output, and all intermediate variables to the users, including the joint kinematics and the muscle fibre lengths during a movement, the neural input to each

```
1   # test motor unit pool model
2   mn_pool = MotoneuronPool(num_mu, rr, rm, rp, pfr1, pfrd, mfr1, mfrd,
    ge, c_ipi, frs1, frsd)          # parameters that define the motor unit
     recruitment and firing pattern
3   # properties
4   config = edict({
5       'num_fb': 25000,
6       'depth': [20, 30],
7       'angle': [20, 30],
8       'iz': [0.5, 0.1],
9       'len': [0.5, 0.1],
10      'cv': [4, 0.5]
11  })
12  properties = mn_pool.assign_properties(config, True)
13  # define neural input
14  fs = 2048           # Hz
15  duration = 6        # s
16  ext = np.linspace(0, 1.0, fs * duration)
17  # Force and twitches
18  mn_pool.init_twitches(fs)
19  mn_pool.init_quisistatic_ef_model()
20  # spike trains
21  _, spikes, fr, ipis = mn_pool.generate_spike_trains(ext)
```

**Code Block 2.** Example python codes for defining a motor unit pool model and for setting the neural input to the pool.

motoneuron pool, the neural commands to the muscles in the format of spike trains, the dynamically changing MUAPs, and the interference surface EMG signals. Given the above information, NeuroMotion can be used to analyse the impact of a specific parameter on the EMG signals and to provide synthetic datasets for data augmentation and validations of EMG-related algorithms. Here we displayed the variations of MUAPs during dynamic contractions and compared the similarities between MUAPs within and across muscles. A case study was provided in which the synthetic dataset generated by NeuroMotion was used to augment an experimental dataset for improving the accuracy in estimating joint angles from EMG signals.

### Synthetic MUAPs during dynamic contractions

**MUAPs when one, two, and three parameters change.** NeuroMotion allows researchers to estimate the changes in muscle fibre length, conduction velocity, and motor unit depth during the movement of the ARMs model (Sections MSK model with OpenSim and Estimate parameter changes). Here, we gave examples of the simulated MUAPs during a wrist flexion and extension movement (Fig 4A) with four levels of parameter changes, including changing only muscle fibre length, changing muscle fibre length and conduction velocity, changing muscle fibre length and motor unit depth, and changing all three parameters. One representative MUAP from the ulnar head of Flexor carpi ulnaris (FCU(u)) and its changes over time are displayed in Fig 4B. By changing the muscle fibre length, the duration of the MUAPs slightly varied. Changing the conduction velocity also influenced the duration of the MUAP while changing the motor unit depth had a more substantial impact on the waveforms. Changes due to the three physiological parameters during the movement are visualised in Fig 4C, 4D and 4E, respectively.

**MUAPs across poses.** We then proceeded to study how MUAPs from different muscles change across movements. Three basic movements were simulated by using the tools described in Section Define movement, including sequences of hand-open to hand-grasp, hand-open to wrist flexion and extension, and hand-open to radial and ulnar deviations. We

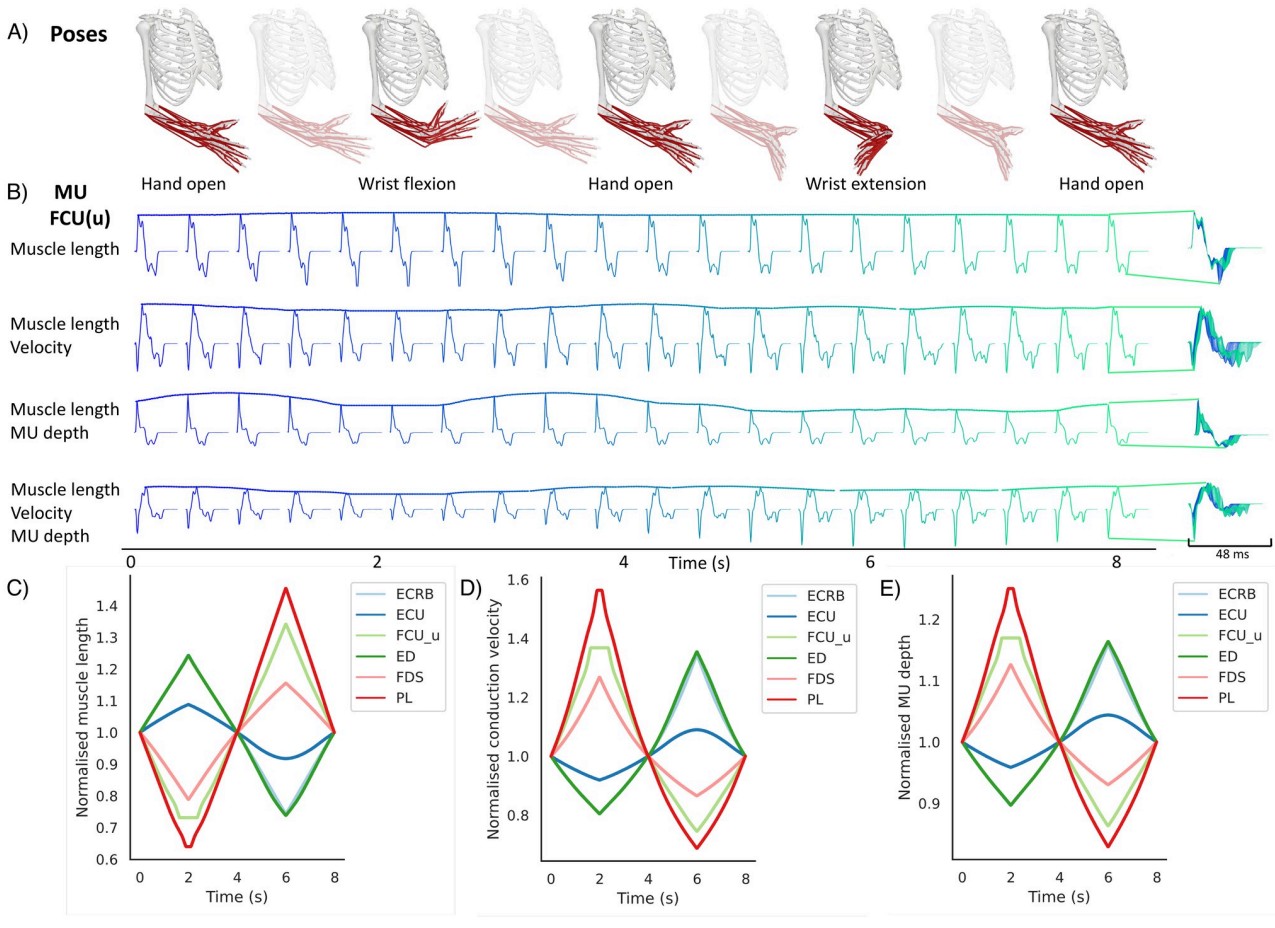

**Fig 4. Changes in the MUAPs of one representative motor unit in FCU(u) during a wrist flexion/extension movement.** A) Sequence of the movement. B) MUAPs generated by NeuroMotion. The MUAPs in the first row were morphed using the muscle fibre length profiles output by the ARMs model using OpenSim, the second row using muscle fibre length and conduction velocity, the third row using muscle fibre length and motor unit depth, and the fourth row using all three parameters. C) Normalised muscle fibre length profiles from the ARMs model. D) Normalised conduction velocity. E) Normalised motor unit depth.

visualised the changes in MUAPs from six superficial muscles. As shown in Fig 5, only the lengths of digitorum muscles (extensor digitorum, ED and flexor digitorum superficialis, FDS) changed during hand grasp and open movement. As a result, MUAPs from ED and FDS displayed large variations while MUAPs from the other muscles maintained their waveforms constant. All six muscles were activated during the flexion and extension movement. Therefore, the duration, amplitude, and waveform of the MUAPs from the six muscles all changed consistently with the movement. The digitorum muscles (ED and FDS) and palmaris longus (PL) muscle contributed less to the radial and ulnar deviations, and thus MUAPs from these three muscles showed fewer variations during wrist deviations.

**In-muscle and cross-muscle MUAP similarities.** Muscles within the forearm have small volumes. Thus, MUAPs within a single forearm muscle might have similar waveforms due to the overlapping motor unit territories. We studied the similarities between MUAPs within a single muscle, across pairs of muscles, and across joint angles during a wrist flexion and extension movement. The similarity was evaluated by the normalised mean square error (NMSE), which is defined by the mean square error between two MUAPs divided by their averaged

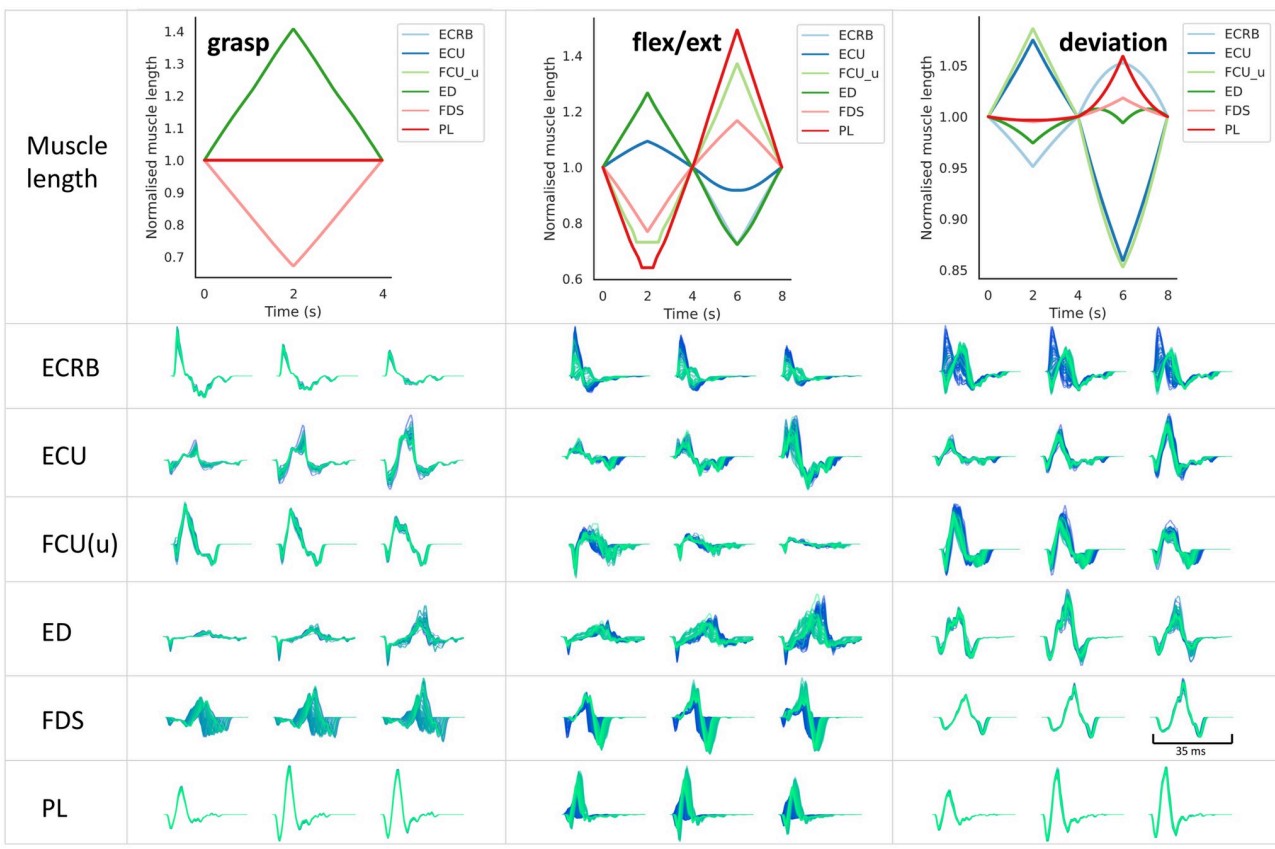

**Fig 5. Changes in MUAPs during three movements.** One representative MUAP was chosen from each of the six muscles and was consistently morphed during three movements. The first row: normalised muscle fibre length profiles from the ARMs model during grasp-open, flexion-extension, and radial-ulnar deviation. The second to the seventh rows: MUAPs transformed during the three movements in the six muscles. The colours from dark blue to light green indicate the time frames from the beginning to the end of the movement. For each MUAP, a subgrid of three channels were chosen for display, which was consistent across the three movements.

power. The MUAPs were first cropped to a sub-grid where all channels had the maximum amplitude above 75% of the average across channels.

The similarities between MUAPs of three muscles, an extensor (Extensor carpi radialis brevis, ECRB) and two heads of one flexor (Flexor carpi ulnaris ulnar and humeral heads, FCU(u) and FCU(h)), are shown by the confusion matrices in Fig 6. MUAPs within the same muscle generally showed higher similarity than MUAPs across different muscles. MUAPs within the two heads of the FCU muscle were more similar than MUAPs within the FCU and a more distant muscle ECRB. Different MUAPs within the same muscle (off-diagonal) could have similar waveforms since multiple combinations of physiological parameters can contribute to similar MUAPs and different MUAPs may overlap in their territories due to the small muscle volume in the forearm.

We were also interested in how movements influence the similarities of the MUAPs and whether the changes in the similarities during the movement are consistent across MUAPs within each muscle. Fig 7 shows the similarities between MUAPs morphed during a wrist flexion/extension movement and their baseline shapes (MUAPs at zero joint angles). MUAPs were gradually morphed away from their baseline shapes when the joint flexion or extension angle increased. The similarities of MUAPs across joint angles were consistent within the

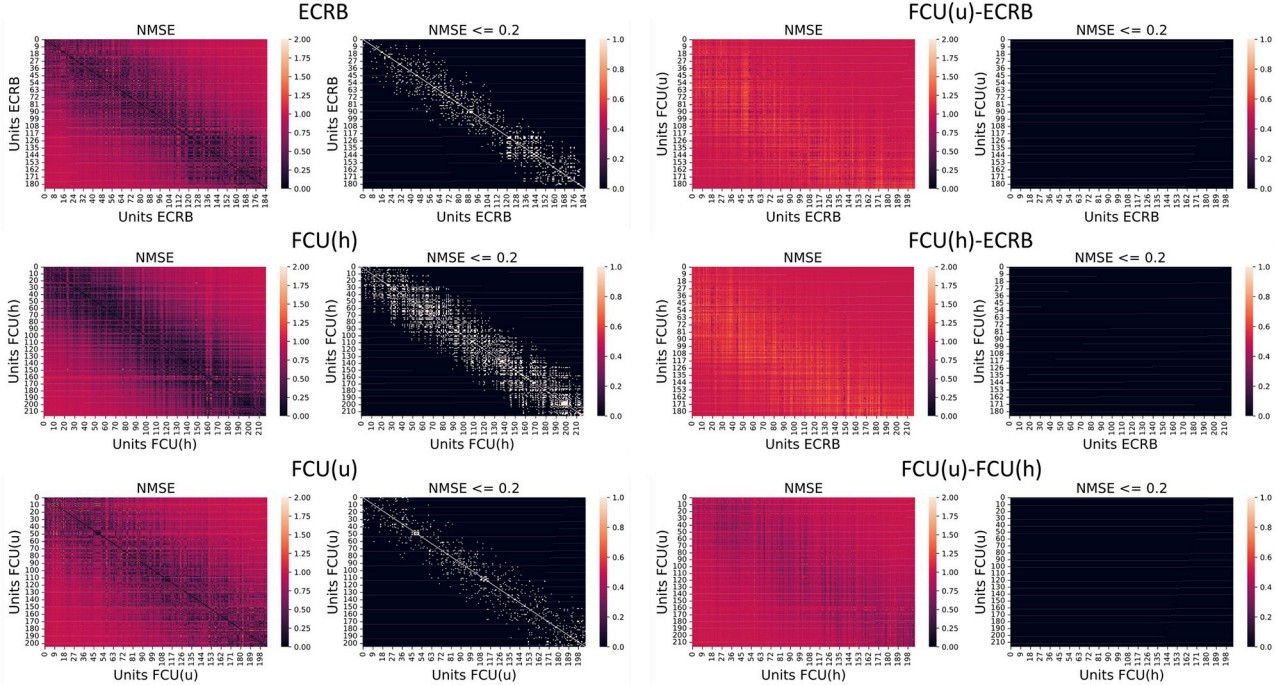

**Fig 6. Similarity and dissimilarity between MUAPs within/across muscles in confusion matrices.** Deeper colours in the first and the third columns indicate a lower normalised mean square error (NMSE) and higher similarity. Black denotes NMSE > 0.2 (less similar) and white NMSE < 0.2 (more similar) in the second and the fourth columns. The left two columns show similarity between MUAPs within single muscles while the right two columns show similarity between MUAPs across two muscles.

MUAPs of each muscle with small variances. MUAPs in extensors showed higher levels of dissimilarity during wrist extension than during wrist flexion, and the opposite for MUAPs in flexor muscles.

## Synthetic EMG signals for data augmentation

**Simulation process.** The synthetic EMG signals generated by NeuroMotion can be used to augment experimental datasets for myoelectric control. In this preliminary case study, we simulated hand and wrist movement following the experiments in [40]. In the experiments, six subjects performed two motor tasks with self-selected speeds, including wrist flexion/extension and MCP flexion/extension movements. Five trials were performed for each task. The duration of each trial was 15 seconds. Six EMG sensors (Delsys Trigno Wireless, Delsys Inc.) were placed on the six forearm muscles according to SENIAM guidelines [41]. The six muscles include flexor carpi radialis (FCR), flexor digitorum superficialis (FDS), flexor carpi ulnaris (FCU), extensor carpi ulnaris (ECU), extensor digitorum (ED), and extensor carpi radialis longus (ECRL). Proper positioning of each EMG sensor was chosen by physically palpating the muscle during sustained isometric contraction and visually confirming the EMG signal. The surface EMG signals were recorded at 2000 Hz and the wrist and MCP joint angles at 40 Hz.

The EMG signals were normalised by the maximum EMG signals during maximum voluntary contractions and then used as the neural inputs to the corresponding muscles to produce spike trains. The joint angles were used to drive the ARMs model, where the output muscle fibre lengths were imported to BioMime to simulate dynamic MUAPs. Note that the latent representation was initialised for each motor unit by randomly sampling from the Normal

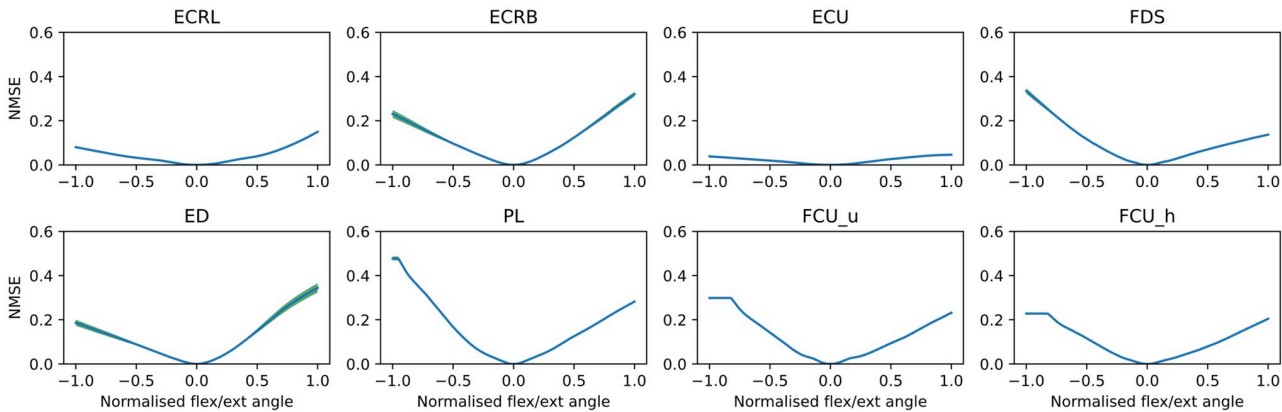

**Fig 7. Similarity between MUAPs across joint angles during a wrist flexion/extension movement.** The joint angles were normalised between -1.0 (flexion) and 1.0 (extension). MUAPs at zero flexion/extension were used as the baseline shapes, which were compared with the continuously morphed MUAPs during the movement to calculate the NMSE (dissimilarity metric, higher values indicate higher differences). NMSEs were averaged within each muscle at each time step, with the variances shown by the green shaded area.

distribution and then fixed during the simulation. This ensures that the MUAPs are only changed by the physiological parameters during the movement. Finally, the synthetic EMG signals were simulated by convolving the spike trains and the MUAPs. We simulated a total of 320 channels of surface EMG for all trials of the six subjects. The simulated channels are uniformly placed around the forearm and cover the majority of the forearm muscles. Six channels were selected for each subject by matching the coordinates between the experimental electrodes and the simulated electrodes. We also checked the amplitudes of the selected channels during specific movements. For instance, the signals collected by channels placed on the flexors should have a higher amplitude during wrist flexion compared to the amplitude during wrist extension. Root mean square (RMS) values were computed from 200-ms intervals with 50-ms overlapping. Each RMS segment was labelled with the two joint angles.

**Case studies of data augmentation.** We investigated whether the simulated signals have practical uses, for example, to augment the experimental dataset in this case study. The assumption here is that even though the modules used in the simulation were not customised to each subject, given the common patterns of muscle activation during movements, the simulated EMG signals could capture the general characteristics of each movement that would be useful for regressing the joint angles. We show that the synthetic EMG signals are of practical use in two ways.

First, we trained ridge regressors on the synthetic dataset from each trial of each subject. The trained regressors were directly applied to regress the joint angles from the experimental data of the same subject and the same trial. We observed that for each of the subjects, there was at least one trial, in which the joint angles predicted by the regressors trained on the synthetic data were highly consistent with the experimental joint angles. The Pearson correlation coefficients (PCC) are 0.52 ± 0.14 (mean and standard deviation across subjects) and 0.61 ± 0.14 for regressing the wrist and MCP angles, respectively. Taking one subject as an example, we found consistent regression results when the simulation was repeated five times. As might be expected, we found that the data produced by NeuroMotion closely matched the real data when the regressors trained on the synthetic data performed well on the real data. An example of the regression results is shown in Fig 8. To account for the fact that the joint angles mainly consist of low-frequency components, we estimated a lower bound of such predictions by comparing the predictions with the randomly shuffled ground truth labels. An extreme case

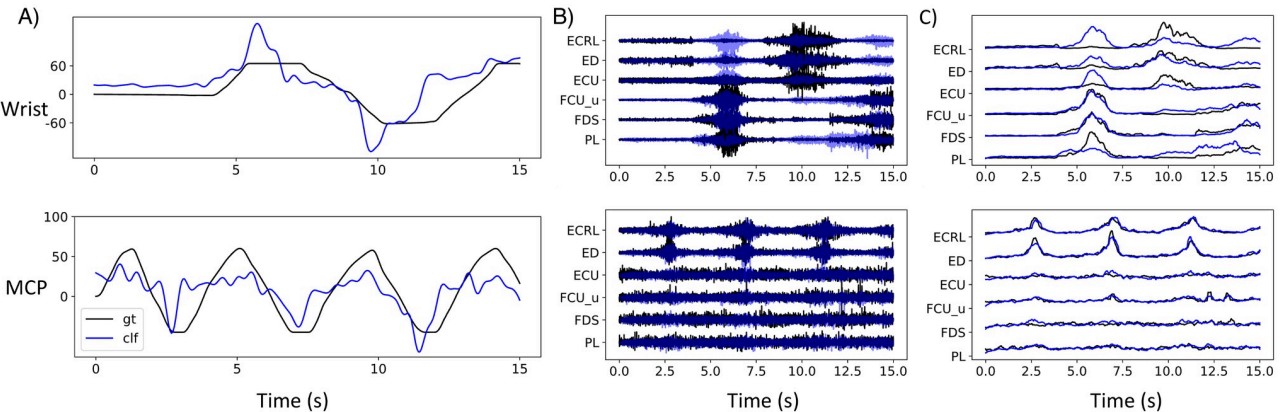

**Fig 8. Representative results of using NeuroMotion's output to train regression algorithms.** A) The regressors trained on the synthetic dataset could predict wrist and MCP joint angles from the experimental data. The ground truth joint angles are displayed in black and the predictions in blue. B) The simulated six-channel EMG signals (blue) and the experimental signals (black) of the same trials in A. C) the RMS extracted from the simulated signals (blue) and the experimental signals (black) of the same trials in A.

is that if the joint angles are constant, shuffling the labels of joint angles will not affect the prediction accuracy. The random shuffling was repeated ten times for each trial. The PCC between the prediction and the randomly shuffled angles has a large variance and contains both negative and positive values, resulting in an averaged PCC around zero ($-0.14 \pm 3.68$). The PCC between the predictions and the ground truth joint angles is significantly higher than the PCC between the prediction and the randomly shuffled angles following the Student's t-test ($p < 0.005$).

Second, we show that the synthetic data can be potentially used to augment the experimental data to improve the regression accuracy. Specifically, we randomly selected two trials of experimental data and two trials of synthetic data to form a new training dataset. The ridge regressors trained on this augmented dataset were tested on one trial of the experimental data, which was different from the trials in the training dataset. For all six subjects, we were able to augment the experimental data with data generated by NeuroMotion such that the resultant regressor performed better than a regressor trained on the experimental data alone. The performance of the regressor was significantly improved when trained on these augmented datasets (Student t-test with $p < 0.05$). The PCC of training on the augmented dataset versus training on the full experimental dataset is $0.77 \pm 0.08$ versus $0.75 \pm 0.08$ for wrist and $0.72 \pm 0.06$ and $0.66 \pm 0.07$ for MCP.

## Discussion

We have presented NeuroMotion, which provides the first open-source simulator for neuromechanical investigations by modelling the electrical outputs during voluntary human movements. With the three key modules, NeuroMotion provides a wealth of functions that give full freedom to users to simulate signals by using the default settings or by customising the configurations. All input, intermediate, and output variables are available in this hierarchical and full-spectrum simulation, indicating multiple potential usages of NeuroMotion.

### Rich variances of MUAPs during dynamic contractions

NeuroMotion provides three ways to define the movement of the ARMs model. The easiest way is to interpolate between (the combinations of) the eight default poses. Examples in Figs 4

and 5 showed simulations of MUAPs by using the interpolation tools. Physiological parameters were smoothly changed during the movement, which resulted in a continuous variation of MUAP waveforms. The variations of synthetic MUAPs are similar to the variations of MUAPs observed during experiments in a qualitative way [42–44]. For example, the shortening of muscle fibres or increase of conduction velocity during natural muscle contractions reduced the duration of a MUAP; MUAPs of muscles that contribute less to a movement showed fewer variations (PL in wrist deviation).

Compared with the limited muscles studied under a few discretised joint angles during experiments, NeuroMotion can simulate MUAPs from eight forearm superficial muscles under any voluntary movements in a high temporal resolution. Such functionality allows users to test the assumptions made in the simulations (e.g., changes in parameters) or study the rich variances of MUAPs within or across muscles as displayed in Fig 6. The ability of NeuroMotion to generate dynamic MUAPs consistent with muscle functionalities demonstrates its superiority to previous models that either adapted the classical cylindrical model to discretised stages [45] or empirically transformed the MUAP waveforms [46]. We expect that NeuroMotion will meet the demand for providing synthetic EMG signals during dynamic muscle contractions for validating adaptive decomposition algorithms [42]. For example, NeuroMotion was used in [47] to generate data during wrist flexion and extension at constant forces. The simulated data were used to test the online learning EMG decomposition algorithm.

We also showed that the generated MUAPs were muscle-specific with low similarity across different muscles (Fig 6). The variations of MUAPs during a movement were consistent within muscle but different across muscles (Fig 7). When the wrist flexion/extension angle increased, MUAPs were gradually transformed and deviated away from their initial version, leading to an increasing difference (higher NMSE). The similarity at each joint angle showed small variances across MUAPs within each muscle, which indicates that deformations of fibres within one muscle are similar. MUAPs from flexors experienced more variations during flexions while MUAPs from extensors showed more variations during extensions. This observation fits well with the fact that flexors/extensors manifest a more drastic contract during flexions/extensions. These results evidence the potential of using NeuroMotion to investigate the variations of MUAPs within/across muscles during voluntary movements.

## Potential as a test-bed for cause-effect studies and data augmentation for regression analysis

With access to all inputs, outputs, and internal parameters in NeuroMotion, users can study cause-effect relationships between the joint kinematics, neural inputs, physiological parameters, muscle forces, MUAPs, spike trains, and surface EMG signals. As a simple demonstration of such studies, we showed how MUAP changes differently with one to three physiological parameters changed during a wrist flexion/extension movement (Fig 4). Coupled with experimental MUAPs observed at the same muscle under discretised movements, such simulations will help test the assumptions made in Section Estimate parameter changes and elucidate the contributions of the physiological parameters to the MUAPs. Another example displays the transformations of MUAPs under three wrist-and-hand movements (Fig 5). This allows the investigation of how MUAP changes when the commonly used forearm DoFs are activated, which will further provide insights into designing and improving current adaptive decomposition algorithms.

NeuroMotion can also be used to study the "inverse" relations between the outputs and the inputs of the system, for example, to pre-train algorithms that regress joint angles from EMG signals and to augment the experimental dataset. Here we provide a preliminary study to show

that ridge regressors trained on the synthetic dataset can be potentially applied to regress joint angles from experimental signals. Furthermore, we observed that the performance of some regressors could be improved when trained on the augmented dataset. These findings are consistent with the perspective of using synthetic datasets to augment the limited actual dataset and facilitate algorithm training [10, 48]. One important observation is that the effectiveness of data augmentation is correlated with subjects. This is expected, as the BioMime module in NeuroMotion only captures the geometry of one specific anatomy from the database, which may deviate from the anatomy of the specific subjects. We discuss this limitation in the next section in more detail.

We also observed that the variance of the augmentation performance across trials for one subject could be large. One possible reason is that we used the normalised EMG signals recorded on each muscle as the neural input, which largely affects the properties of the synthetic EMG signal. Electrode shifts across trials, misalignment between the electrodes and muscles, and crosstalk due to the volume conduction all make the accurate estimation of neural inputs challenging. Moreover, the perspective that motor neurons are controlled in the group of muscles is being challenged by new experimental findings that motor neurons are controlled in clusters that might distribute across muscles [49]. In this case, it is even more difficult to accurately group motor neurons into clusters and estimate the neural inputs to the clusters. Even though, we found that synthetic signals of some channels (muscles) constantly show a higher correlation with their experimental counterparts. This can be explained by the muscle's location, i.e., some muscles are easier to find by palpation. For example, for channels that collect signals from ED and ECRL, the PCC is above 0.7 for wrist regression and above 0.90 for MCP regression. The PCC is lower for slender muscles that are difficult to separate from other muscles (0.40 for PL muscle for wrist angle regression). All these uncertainties of experimental signals relate to the general reasoning behind building forward models like NeuroMotion. The flexibility provided by NeuroMotion allows the investigation of the actual relationship between the neural input and the joint kinematics, for instance, by adjusting the neural inputs to make the synthetic EMG signals more similar to the experimental data under the same protocols. This will in turn give insights into the real relations between joint kinematics and neural inputs during voluntary movements.

## Limitations and future directions

As the first integrated open-source generative model to simulate EMG signals during voluntary movements, NeuroMotion still has some limitations that should be noted. With its modular structure, the intrinsic limitations of NeuroMotion come from the three modules, of which the error is trackable but aggregates in the computation of the predicted EMG signals.

First, the current BioMime model in NeuroMotion is used as a replica of a specific volume conductor but not a universal one. Therefore, there are errors introduced when NeuroMotion is used to simulate signals for specific subjects. A related limitation is that the locations of the electrodes are fixed ($10 \times 32$ channels around the forearm) and cannot be specified. This can be addressed by training a subject-specific BioMime with different electrode configurations. Building a universal BioMime model that can be adapted to specific subjects and allows random samplings of electrical potentials on the surface is also in our future work.

Second, the accuracy of the estimated muscle fibre length will affect MUAP generation, as the muscle fibre lengths are the inputs to the BioMime module to indicate physiological parameter changes. Forearm muscle fibre length is very difficult to estimate experimentally, although some algorithms can predict lower limb muscle fascicle length by tracking the endpoints in the recorded ultrasound images [50]. In NeuroMotion, the muscle fibre lengths are

derived from the geometrical muscle-tendon lengths computed with the ARMs model in OpenSim. Under the assumption that the tendons are rigid and at constant slack length [51], the method does not account for the dynamic changes in muscle fibre lengths to achieve the physiological force equilibrium between the muscle and the extensible tendon. The simplification of neglecting the compliance of tendons, taken for computational convenience and modelling simplicity, may be a source of error when estimating muscle fibre lengths, as quantified in [51], and discussed for forearm muscle-tendon systems in [52].

Third, the accuracy of the spike trains estimated from the Motor Unit Pool module is affected by both the neural inputs to the module and the module itself. The neural inputs estimated from the muscle activations computed by the OpenSim's Static Optimisation tool are sensitive to the aforementioned inaccuracies in muscle fibre length [13] and the intrinsic limitations of Hill-type muscle modelling [53]. This includes multiscale modelling simplifications [54], physiological and numerical instabilities [55], and limited phenomenological modelling of the muscle's activations dynamics [53, 56, 57]. The Static Optimisation method also computes muscle activations, and hence, neural inputs without considering the concurrent dynamics of the proprioceptive, spinal and propriospinal circuits and reflexes during voluntary movement. These concurrent dynamics can be modelled and were demonstrated to have effects in the simulation of neuromuscular and musculoskeletal function [58–63]. The deafferented approach taken in the present study does not consider the sensory feedback related to passive changes of fibre length, which would result in inaccuracies in the estimated muscle fibre lengths and muscle activations and thus simulated EMG as well. The EMG-informed estimation of neural inputs (used in Section Synthetic EMG signals for data augmentation) is sensitive to the normalisation and the experimental settings, for example, the electrode placement and crosstalk [64]. Inaccuracy in the estimated spike trains also comes from the limitations of motor unit pool models. The two models used in NeuroMotion are designed for isometric contractions and do not consider changes in motor neuron discharge patterns associated with voluntary movement [16, 29, 44]. These models are also not complex enough to physiologically describe the intrinsic discharge properties and mechanisms of motor neurons (e.g., the nonlinear transformation of common synaptic inputs into total dendritic membrane current discussed in [65]).

Even with all the limitations of each module, the modular structure of NeuroMotion allows it to be improved with the emergence of new measuring and modelling approaches in the EMG field. For example, *in vivo* measurements of muscle activations, muscle fibre length, and other physiological parameters would contribute to a more accurate estimation of neural inputs and MUAPs. More complex modellings of motor neuron pools could account for nonlinearities in spiking dynamics and afferent feedback activities. We would expect NeuroMotion to be frequently updated with the advancements in each module and be validated in future by comparing the synthetic signals with the experimental data.

## Conclusion

We proposed NeuroMotion, an open-source EMG simulator that generates surface EMG signals during human forearm movements. For the first time, NeuroMotion provides a core resource for the neuromechanics community to address the problem of simulating physiological electrical outputs during biomechanical movements and allows a full-spectrum simulation from movements to neural commands and EMG signals. All intermediate variables (kinematics, dynamic MUAPs, spike trains) are available during the simulation, which makes the model flexible to use for the users' purposes. We demonstrated that one potential application is to use synthetic data to augment the experimental data for training regression algorithms.

We would expect that with the functionality and extensibility provided by NeuroMotion, users can customise the way they utilise NeuroMotion that will ultimately yield progress in the neuromechanics fields.

## Author Contributions

**Conceptualization:** Shihan Ma, Dario Farina.

**Data curation:** Shihan Ma, Jiamin Zhao, Kostiantyn Maksymenko, Samuel Deslauriers-Gauthier.

**Formal analysis:** Shihan Ma, Dario Farina.

**Funding acquisition:** Xinjun Sheng, Xiangyang Zhu, Dario Farina.

**Investigation:** Shihan Ma, Irene Mendez Guerra, Arnault Hubert Caillet.

**Methodology:** Shihan Ma, Arnault Hubert Caillet.

**Project administration:** Xinjun Sheng, Xiangyang Zhu, Dario Farina.

**Resources:** Shihan Ma, Dario Farina.

**Software:** Shihan Ma.

**Supervision:** Dario Farina.

**Validation:** Shihan Ma.

**Visualization:** Shihan Ma.

**Writing – original draft:** Shihan Ma.

**Writing – review & editing:** Shihan Ma, Irene Mendez Guerra, Arnault Hubert Caillet, Alexander Kenneth Clarke, Dario Farina.

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
