## [Decision Letter · Decision Letter 0]

2 Feb 2024

Dear Prof. Farina,

Thank you very much for submitting your manuscript "NeuroMotion: Open-source Simulator with Neuromechanical and Deep Network Models to Generate Surface EMG signals during Voluntary Movement" for consideration at PLOS Computational Biology.

As with all papers reviewed by the journal, your manuscript was reviewed by members of the editorial board and by two independent reviewers. The reviewers agreed that the paper presents an impressive and potentially important synthesis of previous models. However, they both agreed that the validation of the model is insufficiently rigorous at present. In addition, they felt that potential limitations of the model (e.g. related to the specific modeling assumptions) should be more prominently discussed.

We would like to invite the resubmission of a significantly-revised version of the paper that addresses these important concerns, as well as other concerns raise by the reviewers. We cannot make any decision about publication until we have seen the revised manuscript and your response to the reviewers' comments. Your revised manuscript is also likely to be sent to reviewers for further evaluation.

Sincerely,

Adrian M Haith

Academic Editor

PLOS Computational Biology

Daniele Marinazzo

Section Editor

PLOS Computational Biology

Reviewer's Responses to Questions

**Comments to the Authors:**

Reviewer #1: The authors proposed an innovative electromyographic generative model aimed at reproducing electrophysiological signals (BIOMIME) during dynamic contractions. The new model includes a previously developed model for the generation of motor unit action potentials during dynamic contractions, an OPENSIM muscular skeletal model for the simulation of muscle morphological parameters of the upper limb during dynamic movements, and a motor unit pool model to generate the spike trains. In general, the manuscript is well written and extremely important for the field, but the authors should improve the justification for their work. In particular, the innovation of the present work is not clear in comparison to the state-of-the-art. Additionally, some of the conclusions require further validation. Specifically:

• Validation results are reported for three subjects only, and no statistical analysis has been performed. The synthetic EMG signals have been tested to predict the joint angles of three subjects. The number of subjects is very limited, and the authors reported only the averaged correlation coefficients across different movements, making it difficult to evaluate the performance of the model. Specifically, it is complex to evaluate how good a PCC of 0.54 or 0.72 is without a comparison to anything else. These statistics also refer only to one subject, while for the other two, only the best regressors are mentioned, not the average among all DoF. A larger pool of subjects should be used, and a full statistical analysis of the results (average, std, distribution of the results, etc.) across all subjects should be presented. For example, in the manuscript, no information is presented about the results on the third subject, and in this specific case, it is rather important since it constitutes 1/3 of the validation dataset.

• Since the generation of the synthetic dataset is non-deterministic, multiple datasets could also be generated and tested, and the authors should check the consistency of different simulations.

• The results of the proposed model have not been compared to other state-of-the-art methods or models.

• The proposed model has been validated on a subset of six channels only. It is not clear how these six channels have been selected from the 320 generated from the model and how they were matched with the positions on the muscle of the experimentally recorded EMG channels.

• How were the experimental electrodes positioned on the muscles? Previous studies have shown that small misalignments between the position of the electrodes and the location of the individual muscles may create large errors in the performance of the regression.

• The proposed model should be validated using a high-density dataset in combination with motor unit decomposition to compare the simulated variation in MUAP shapes with the ones estimated experimentally.

• Figure 8 reported the comparison between the regressions performed with the simulated and experimental signals. However, the comparison should be performed using multiple repetitions and subjects.

• Joint angle signals have very slow frequency components; therefore, the correlation coefficients with such slow signals may have a quite large lower bound. For these reasons, the authors should shuffle their dataset to estimate this lower bound and demonstrate that their results are significantly higher than that confidence level.

Reviewer #2: Review of

NeuroMotion: Open-source Simulator with Neuromechanical and Deep Network Models to Generate Surface EMG signals during Voluntary Movement

Ma et al.

This paper proposes to create synthetic EMG that is compatible with muscle mechanics and motor unit function, taking into account the volume conduction of activation patterns through tissue on its way to the electrodes. As such, it is a very good first attempt towards generation of EMG in simulation, which has not yet been validated against a ground truth.

Major Comments

1. Figure 1: Based on Figure 1, the authors synthesize the EMG signals for the mentioned signals using the outputs from the simulated motion in OpenSim.

○ Have the authors examined the difference between the EMG signals recorded experimentally in the real world and the synthesized signals? If yes, how different are these signals? What are the metrics to study these differences?

○ What is the neural drive? Line 289-311 code block 2

○ What are the “other parameters”.

2. Muscle models: The muscle models used by OpenSimm and the Fuglevand model, Leaky Integrate and Fire (LIF) have multiple known limitations. For example, a recent review by Herzog emphasizes those for the Hill-Type model.

https://royalsocietypublishing.org/doi/pdf/10.1098/rsif.2022.0430

While the authors are providing a computational, modular tool that can be repurposed, it is critical that the authors mention this as a strong limitation of their results.

3. The system at this point is not compared to ground truth data, which is an important and critical limitation that needs to be addressed.

4. More importantly, there is a critical assumption that is problematic: Muscles are considered de-afferented and only responsive to the neural drive (muscle activation). Such proprioceptive, spinal and propriospinal circuits greatly affect the activity of the motor units. This needs to be mentioned and highlighted as a critical limitation.

5. Similarly, it is important to highlight the critical limitation of how to handle passive lengthening and shortening in real limbs when the source neural drive to a muscle is not known.

6. The points made in 1—5 above are critical to the proposed use as “Test-bed for Cause-Effect Studies and Data Augmentation for Regression Analysis”. This needs to be mentioned and the utility for this purpose toned down at this point.

7. Lines 155-157: As the text mentions, the toolbox incorporates two distinct neuron models (the classical Fuglevand's model and Leaky integrate and fire LIF). This applies to the Hill-type model as well.

○ Conducting a comparative analysis of results obtained from each model module could offer valuable insights into their respective performance and applicability. For example, a simple Izhikevich neuron is a popular alternative to the LIF.

○ Additionally, it would be beneficial to delve into more nuanced considerations regarding the practical utility of each model module, providing a comprehensive exploration of their specific use cases, advantages, and potential limitations.

○ Such an examination would not only facilitate a deeper understanding of the toolbox's capabilities but also guide users in selecting the most suitable neuron model based on the specific requirements of their research or applications.

8. Line 147: The authors mention that the bulk of variations of the MUAPs are explained by the seven parameters; what is the basis for that? Is there a measurement or reasoning behind that?

9. BIOMIME: Figure 2: The authors are using an AGNN in order to predict the MUAPs. The AGNN is conditioned to specific parameters.

○ First, are the specified conditions the same as the seven supported parameters? If yes, do you train the pre-trained model using the specific conditions from the subject?

○ In general, do you train the model at all? Or you just feed the samples and condition into the model for the inference mode?

○ The figure is incomplete as it does not show how the signals of the discriminator are used.

10. Lines 365-367: Authors are showing the similarity of the MUAPs in one muscle and the differences with the ones from other muscles. But this is not surprising, simply a quality control step, Figure 6.

○ What is the point that is being made? To me, the important issue to discuss is the source of off-diagonal signals in the within-muscle comparison.

Minor comments:

11. Line 98: Assuming tendons to be rigid at constant slack length in musculoskeletal models can lead to several errors, including inaccurate force transmission, misrepresentation of muscle length changes, incorrect muscle activation patterns, inadequate representation of energy dynamics, and limited accuracy in predicting joint kinematics. It could be helpful to mention the errors introduced by this assumption.

12. Line 141: You mention that you are only using the generator (as it is in inference mode), however, the discriminator plays a pivotal role in assessing the realism of synthetic data compared to real counterparts. The discriminator's confidence scores, ROC and precision-recall curves, confusion matrix, and standard statistical metrics offer comprehensive insights into the model's ability to distinguish between real and synthetic data. Leveraging these evaluation methods enables a thorough understanding of the generator's performance, facilitating continuous monitoring and refinement to ensure the generation of high-quality synthetic data.

13. Line 149: Authors mention that the toolbox supports both methods, is there any comparison between the two of them?

**Have the authors made all data and (if applicable) computational code underlying the findings in their manuscript fully available?**

Reviewer #1: None

Reviewer #2: Yes

PLOS authors have the option to publish the peer review history of their article (what does this mean?). If published, this will include your full peer review and any attached files.

Reviewer #1: No

Reviewer #2: No
---

## [Decision Letter · Decision Letter 1]

31 May 2024

Dear Prof. Farina,

Thank you very much for submitting your manuscript "NeuroMotion: Open-source Simulator with Neuromechanical and Deep Network Models to Generate Surface EMG signals during Voluntary Movement" for consideration at PLOS Computational Biology. As with all papers reviewed by the journal, your manuscript was reviewed by members of the editorial board and by several independent reviewers.

Overall, the reviewers appreciate the methodological advance provided by NeuroSim, and believe that it is a very valuable contribution to the field. The one remaining concern from Reviewer 2, however, is that the present capabilities may be slightly oversold. There remain some substantial limitations and, although these are well covered in the Discussion, the impression conveyed in earlier parts of the paper is that the problem has now been definitively solved. I do not believe this is a major issue, but I agree with Reviewer 2 that it is important to more clearly convey that there are limitations to the present approach, and that this line of research remains a work in progress for which further improvements and refinement are expected in the future (while taking nothing away from the fact that this is a very successful and impressive first implementation of a model along these lines, and will already be of substantial value).

Sincerely,

Adrian M Haith

Academic Editor

PLOS Computational Biology

Daniele Marinazzo

Section Editor

PLOS Computational Biology

Reviewer's Responses to Questions

**Comments to the Authors:**

Reviewer #1: The authors have done an excellent work in revising the manuscript. All my previous comments have been addressed.

Reviewer #2: Review of

NeuroMotion: Open-source Simulator with Neuromechanical and Deep Network Models to Generate Surface EMG signals during Voluntary Movement

Ma et al.

I thank the authors for their detailed responses to the reviewers. I will leave it to R1 to assess the completeness of the revised version their comments. Considering those comments and mine (R2), I have some additional clarifications, and in the interest of time, I will only mention major ones.

Limitations of this study:

Many of the concerns that were expressed were addressed in the methods and discussion. In addition, the revised MS now has a more comprehensive expanded limitations section that includes many of those issues. However, it is worth noting that the limitations are multiple and important.

1. So, at the end of the day, does the evidence support the claims made in the title and abstract, which were not revised? After the great effort made by the reviewers and authors to create the revised version, my perspective is that, unfortunately, the evidence does not entirely support the claims made in the original manuscript. This is amply elaborated in the reviewer’s comments and revised Limitations.

2. The value of this work is the platform. The validity of the muscle physiology and EMG generation that are used in this first use-case are, as per the limitations, debatable and simply one of the options the field currently has. The authors need not limit the value of their work to the initial generative EMG work presented (which has good points but also many limitations). There is no need for the value of the work to be dependent on the details of this first use-case.

3. I have much respect for the authors so, it is with humility and solely in the interest of time, that I point out what I believe the evidence supports (tracked changes as CAPS).

a. Original Title (not edited in the revised version): “NeuroMotion: Open-source Simulator with Neuromechanical and Deep Network Models to Generate Surface EMG signals during Voluntary Movement”.

b. My commented one “NeuroMotion: Open-source PLATFORM with Neuromechanical and Deep Network MODULES to Generate Surface EMG signals during Voluntary Movement

This is important because the main and well-justified contribution is the modular computational platform. As to the results and validity of the specific initial EMG predictions, the Limitations listed clearly show this is simply a first use-case.

Abstract (not edited in the revised version)

4. Original text “…However, current simulation models of electromyography (EMG), a core physiological signal in neuromechanical analyses, are either limited in accuracy and conditions or are computationally heavy to apply. Here, we overcome these limitations by presenting NeuroMotion, an open-source simulator that allows a full-spectrum synthesis of EMG signals during voluntary movements….”

5. My suggested text: “…However, current simulation models of electromyography (EMG), a core physiological signal in neuromechanical analyses, REMAIN either limited in accuracy and conditions or are computationally heavy to apply. Here, we PROVIDE A COMPUTATIONAL PLATFORM TO ENABLE FUTURE WORK TO overcome these limitations by presenting NeuroMotion, an open-source simulator that CAN MODULARLY TEST A VARIETY OF APPROACHES TO THE full-spectrum synthesis of EMG signals during voluntary movements….”

6. Original text: “… NeuroMotion is comprised of three modules.”

7. My suggested text: “WE DEMONSTRATE NeuroMotion USING three SAMPLE modules.”

8. Original text: “…NeuroMotion is the first full-spectrum EMG generative model to simulate human forearm electrophysiology during voluntary hand, wrist, and forearm movements. ….”

9. My suggested text: “…IN THIS WAY, NeuroMotion WAS ABLE TO GENERATE full-spectrum EMG FOR A FIRST USE-CASE of human forearm electrophysiology during voluntary hand, wrist, and forearm movements. ….”

10. Original text: “…We expect this full-spectrum model will complement experimental approaches and facilitate neuromechanical research.”

11. My suggested text: “…We expect this MODULAR PLATFORM will ENABLE VALIDATION OF GENERATIVE EMG MODELS, complement experimental approaches and EMPOWER neuromechanical research.”

These edits to the Abstract are super critical as the abstract needs to reflect the main defensible (and valuable) computational contributions, and (implicitly) acknowledge the many limitations at the physiological levels.

Making these changes—**and toning down the claims in the Discussion to align with this**—allows for a very valuable MS without the need to do many, if any, revisions to the Methods or Results.

12. In the modeling community, many are now moving away from OPENSIMM un favor of using MuJoCo as the physics and graphics engine. It is a more computationally efficient platform that reads SIMM-format XML files. MuJoCo may, incidentally, be a better platform for the computationally demanding approach the authors use. Readers should be made aware of this movement in the field. For example, MyoSim

Wang, H., Caggiano, V., Durandau, G., Sartori, M. and Kumar, V., 2022, May. MyoSim: Fast and physiologically realistic MuJoCo models for musculoskeletal and exoskeletal studies. In 2022 International Conference on Robotics and Automation (ICRA) (pp. 8104-8111). IEEE.

Please add other citations as well.

13. One of my original comments was the importance of including subcortical circuits. This was just mentioned in the Limitations but with no references.

a. I suggest this change of wording that add reflexes explicitly (“The Static Optimisation method also does not consider the concurrent dynamics of the proprioceptive, spinal and propriospinal circuits AND REFLEXES during voluntary movement in the computation of muscle activations, and hence, of neural inputs.”)

b. References should be included. A good reference is below, but please add others.

Song, S., Kidziński, Ł., Peng, X.B. et al. Deep reinforcement learning for modeling human locomotion control in neuromechanical simulation. J NeuroEngineering Rehabil 18, 126 (2021). https://doi.org/10.1186/s12984-021-00919-y

14. It remains hugely problematic to use muscle fiber length (which can be passive) as a means to estimate EMG—especially when the model of the muscle is deafferented, as per my prior review. This should be mentioned in the limitations with references.

15. So sorry, but in response to one of my comments, the justification for ignoring tendon length and elasticity as “theoretically acceptable” is much too facile. Zajac himself used forearm muscles as an example of where the properties of these long tendons are important. It is OK to make assumptions, and also OK to say they are done in the interest of computational convenience—but there is no need to say it is physiologically correct.

Zajac, Felix E. "How musculotendon architecture and joint geometry affect the capacity of muscles to move and exert force on objects: a review with application to arm and forearm tendon transfer design." The Journal of hand surgery 17.5 (1992): 799-804.

It is my sincere hope that these major comments will allow this work to become a computational cornerstone to help us move the field forward.

**Have the authors made all data and (if applicable) computational code underlying the findings in their manuscript fully available?**

Reviewer #1: Yes

Reviewer #2: Yes

PLOS authors have the option to publish the peer review history of their article (what does this mean?). If published, this will include your full peer review and any attached files.

Reviewer #1: No

Reviewer #2: No

Figure Files:

Data Requirements:

Reproducibility:

References:

---

## [Editor Report · Decision Letter 2]

15 Jun 2024

Dear Prof. Farina,

We are pleased to inform you that your manuscript 'NeuroMotion: Open-Source Platform with Neuromechanical and Deep Network Modules to Generate Surface EMG signals during Voluntary Movement' has been provisionally accepted for publication in PLOS Computational Biology.

Best regards,

Adrian M Haith

Academic Editor

PLOS Computational Biology

Daniele Marinazzo

Section Editor

PLOS Computational Biology

---

## [Editor Report · Acceptance letter]

22 Jun 2024

PCOMPBIOL-D-23-01615R2 

NeuroMotion: Open-Source Platform with Neuromechanical and Deep Network Modules to Generate Surface EMG signals during Voluntary Movement

Dear Dr Farina,

I am pleased to inform you that your manuscript has been formally accepted for publication in PLOS Computational Biology. Your manuscript is now with our production department and you will be notified of the publication date in due course.

With kind regards,

Zsofia Freund
